# Deployable Tubular Mechanisms Integrated with Magnetic Anchoring and Guidance System

**Wenchao Yue** [1,2,3,†] , **Ruijie Tang** [1,†] , **Joei Simin Wong** [2,†] **and Hongliang Ren** [1,2,3,4,*]

1   Department of Electronic Engineering, The Chinese University of Hong Kong, Hong Kong 999077, China; wenchao.yue@link.cuhk.edu.hk (W.Y.); ruijie.tang@link.cuhk.edu.hk (R.T.)
2   Department of Biomedical Engineering, National University of Singapore, Singapore 117575, Singapore; e0201259@u.nus.edu
3   National University of Singapore (Suzhou) Research Institute, Suzhou 215000, China
4   Shun Hing Institute of Advanced Engineering, The Chinese University of Hong Kong, Hong Kong 999077, China
*   Correspondence: hlren@ieee.org; Tel.: +852-3943-8453
†   These authors contributed equally to this work.

**Abstract:** Deployable mechanism has received more attention in the medical field due to its simple structure, dexterity, and flexibility. Meanwhile, the advantages of the Magnetic Anchoring and Guidance System (MAGS) are further highlighted by the fact that the operators can remotely control the corresponding active and passive magnetic parts in vivo. Additionally, MAGS allows the untethered manipulation of intracorporeal devices. However, the conventional instruments in MAGS are normally rigid, compact, and less flexible. Therefore, to solve this problem, four novel deployable tubular mechanisms, Design 1 (Omega-shape mechanism), Design 2 (Fulcrum-shape mechanism), Design 3 (Archway-shape mechanism), and Design 4 (Scissor-shape mechanism) in this paper, are proposed integrated with MAGS to realize the laser steering capability. Firstly, this paper introduces the motion mechanism of the four designs and analyzes the motion characterization of each structure through simulation studies. Further, the prototypes of four designs are fabricated using tubular structures with embedded magnets. The actuation success rate, the workspace characterization, the force generation and the load capability of four mechanisms are tested and analyzed based on experiments. Then, the demonstration of direct laser steering via macro setup shows that the four mechanisms can realize the laser steering capability within the error of 0.6 cm. Finally, the feasibility of indirect laser steering via a macro-mini setup is proven. Therefore, such exploration demonstrates that the application of the deployable tubular mechanisms integrated with MAGS towards in vivo treatment is promising.

**Keywords:** deployable structure; tubular mechanism; magnetic actuation; laser steering; MAGS

## 1. Introduction

Nowadays, minimally invasive surgery (MIS) has become the development trend and direction due to its small incision, short recovery period, and high operational precision [1–3]. Laparoendoscopic single-site surgery (LESS) and natural orifice transluminal surgery (NOTES) are two typical extensions of MIS techniques to treat in vivo diseases [4,5]. To work towards LESS or NOTES, the Magnetic Anchoring and Guidance System (MAGS) was introduced by Cadeddu et al. [6] to reduce the number of instruments occupying the space at the access port. MAGS's potential to overcome the limitations of LESS and restore triangulation is unanimously agreed upon by researchers, as evident from the research on MAGS [6–9]. Intra-abdominal and extracorporeal components are anchored together against the abdominal wall via magnetic coupling forces [8,9]. During surgery, the intra-abdominal component will be inserted into the body through a small incision at the abdominal wall. It will be controlled and navigated around the peritoneal cavity by the extracorporeal component. LESS integrated with a magnetic anchored camera has been implemented on the human body successfully [8]. Furthermore,

depending on the requirements of different surgical applications, the MAGS can be designed to contain surgical instruments such as a camera, grasper, retractor, light source [10,11].

Magnetically actuated robots have been a growing area of interest for many researchers due to their increasing applications in the MIS, including soft continuum robots, soft capsule endoscope robots, and miniature robots [12–14]. The magnetic field is known to have negligible interaction with body fluids and tissues, so it does not impose any adverse effect on the human body [15,16]. Apart from its safety aspect, magnetic actuators possess desirable features for biomedical applications, such as their quick response to input signals, low-cost materials, and the ability for tetherless manipulation in narrow regions [17–19]. Tetherless manipulation of the robots in the intracorporeal environment allows minimally invasive medical procedures to be more dexterous [20].

Deployable structures are a class of structures that can unfold from the collapsed state to the predetermined working state, and the concept was proposed by NASA in the 1960s [21]. When the structure is folded, it has a smaller volume, facilitating transportation and storage [22]. When turning into the unfolded state, the structure has a large envelope volume and working surface, which can meet the functional requirements of MIS (like NOTES, LESS, etc.), and it has shown huge potential in the medical field [23–25].

Extensive research has shown that orientation guidance tasks have been widely explored in magnetic robotic fields. The patterned magnetization is applied to millimeter-scale robots to achieve laser guiding with various locomotion mechanisms such as multi-arm grasping and multi-legged paddle crawling [26]. An external electromagnetic actuation system can control the Capsule endoscopic robot with a biopsy punch to realize the extrusion and retraction motion [27]. Erin et al. designed a capsule endoscopic robot with hyperthermia and drug delivery capabilities that can be applied in MRI procedure [28,29]. The magnetic robots can be used for needle insertion under multi-axis control [30].

This paper explores four novel deployable tubular mechanisms integrated with MAGS. Such proposed designs can help achieve the untethered operations in vivo towards potential applications such as natural orifice transluminal treatment and single-site surgery as shown in Figure 1. To sum up, our contributions to this manuscript are as follows:

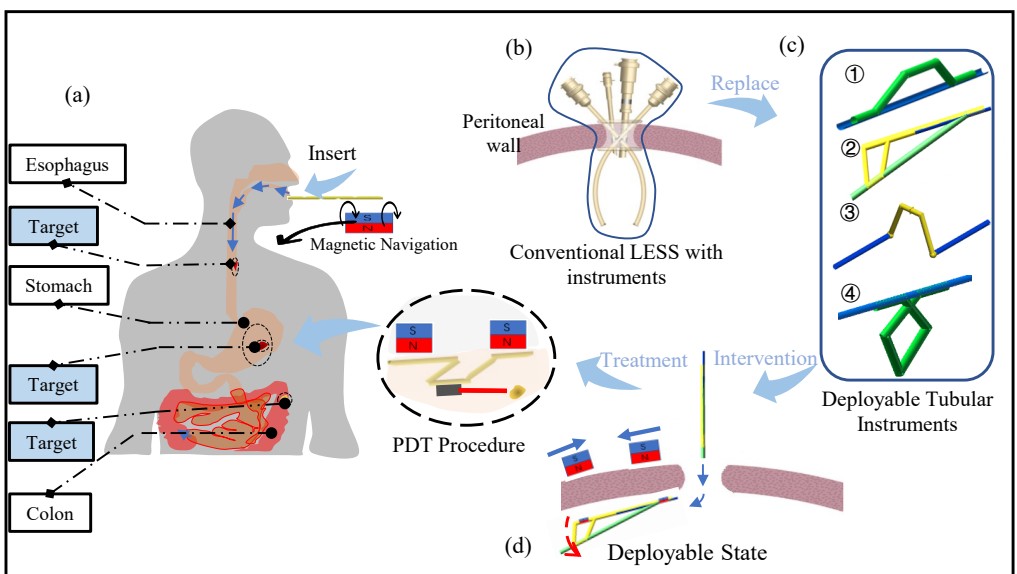

**Figure 1.** (**a**) Potential application of our proposed deployable structure in the gastrointestinal treatment through a natural orifice or small incision. (**b**) traditional laparoendoscopic single-site surgery (LESS) with lots of rigid operation instruments which further limit the workspace and flexibility; (**c**) shows that four types of our proposed deployable structures that can replace the rigid instruments in the conventional LESS, which can further increase the operation efficiency, and also minimize the incision size; (**d**) indicates that when the proposed structure inserts into the abdomen through one single incision, it would be deployed to realize light steering capability in vivo treatment such as photodynamic therapy (PDT).

1.    We propose four deployable tubular mechanisms integrated with MAGS based on the common plastic straws and permanent magnets, which can achieve low cost and fast assembly;
2.    We demonstrate the characterization of four magnetic-driven deployable tubular mechanisms, based on actuation success rate, workspace characterization, load analysis, and force analysis;
3.    We demonstrate the laser steering experiments on different trajectories with a maximum mean error below 0.6 cm, which shows the application potential of such deployable structures integrated with MAGS.

## 2. Materials and Methods

This actuation system is prototyped and analyzed mainly using two materials, namely, drinking straws and neodymium–iron–boron (NdFeB) permanent magnets. The magnets used for designs are listed in Table 1. The prototypes are assembled by means of sticking and cutting using simple blue tack, sticky tape, and thin metal wires. As initial proof-of-concept prototyping material, straws ease the modification by making simple incisions along the straw's length and creating hinge joints. The hinge joints determine the bending direction and kinematic performance of the prototype. Hence, we create different tubular mechanical systems with hinge joints at different positions and orientations along with the straws. The straws utilized for prototyping in this paper have a length of 20 cm and a diameter of 8 mm. Most of the permanent magnets used in MAGS devices or MAGS microrobots are NdFeB magnets [10,11]. This type of magnet has a high magnetic strength-to-volume or weight ratio, and they are not easily demagnetized, making them the strongest grade of magnets available [6]. In terms of actuation distance, the statistical data showed that the average patients have abdominal wall thickness of 2 cm [31]. Our designs are aimed to provide the preliminary verification that can be applied to animal experiments. It is reported using a dual-stack NeFeB magnets weighing 583 g as the external magnet allowed a maximum inter-magnetic distance of 4.78 cm with the heaviest load of 39 g [32]. Therefore, to verify the preliminary designs, we use the prototype consisting of a stack of (NdFeB) permanent magnets with the size of 3.2 cm $\times$ 3.2 cm $\times$ 2 cm placed on the outer abdominal wall (thick 0.7 cm), which can afford the weight of 1000–1500 g regarding its attractive force of MAGS. While the size of the inner magnet is chosen as the cylinder with a diameter of 0.7 cm and a length of 1 cm, which can be easily integrated inside the staw tubes. It is important to note that NdFeB magnets are easily corroded once their surface plating is damaged. Thus, NdFeB magnets should not be in direct contact with the human body's internal surroundings [33]. Hence, they are placed and fixed inside the straws for all the prototypes in this paper.

**Table 1.** The types of magnets used in prototypes.

| Type of Magnets | Size | Br (mT) |
| --- | --- | --- |
| N35 NdFeB | Cuboid: 20 mm $\times$ 10 mm $\times$ 3 mm; | 1210 |
| N38 NdFeB | Cuboid: 20 mm $\times$ 10 mm $\times$ 3 mm; | 1260 |
| N52 NdFeB | Cube: 25 mm | 1450 |

MAGS is originally investigated for its ability to perform simple surgical tasks [34] as Figure 2a shows, to manipulate some functional modules such as a surgical camera [35], surgical retractors [35], and surgical cauterizer [36]. Previous trials highlighted the benefits of having a mobile camera hanging from the abdominal wall compared to the use of conventional endoscopes that suffer from tunnel vision, i.e., misperception of the surgical workspace for surgeons [37]. To solve this problem, deployable structures would be introduced to increase the flexibility and dexterity of surgery tools. Two external PM anchors are arranged such that their relative distance can be varied [8], resulting in the actuation of a device functional unit through a simple mechanism shown in Figure 2b.

Therefore, our work further proposes multiple designs for deployable structure parts to achieve more motion characteristics.

This paper explores four types of prototype with different deployable tubular mechanisms, as shown in Figure 3a–d, and their corresponding actuation demonstrations are indicated in Figure 4. Each of these prototypes has at least two embedded magnets. Thus, the orientation of the embedded magnets is carefully thought out to ensure that magnetic coupling between the internal (or external) magnets does not occur during actuation. Since like-poles repel, the adjacent embedded magnets are orientated in the same direction (North pole along the same direction) to prevent magnetic coupling between the adjacent magnets even when they are approaching close together.

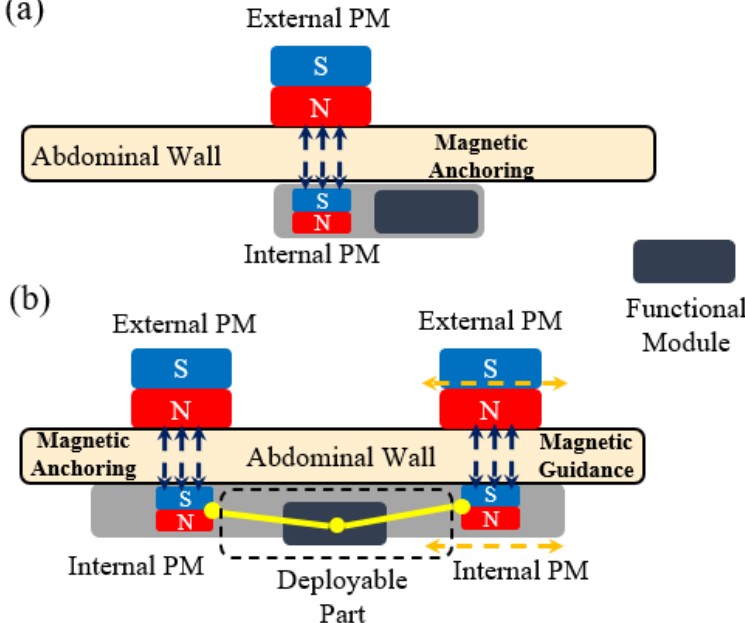

**Figure 2.** (**a**) Conventional rigid function model in MAGS. (**b**) Deployable structure actuated by the two external Permanent Magnets (PM).

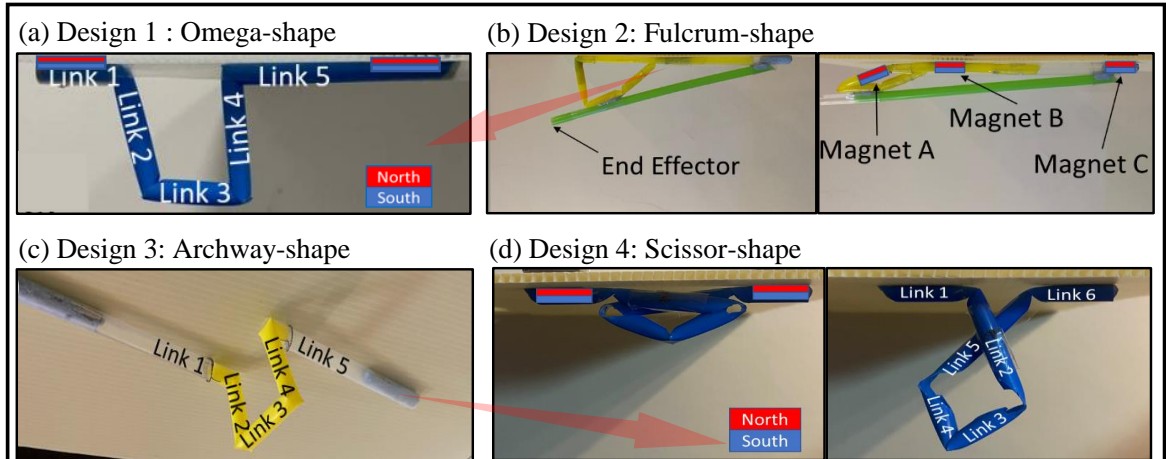

**Figure 3.** (**a**) shows the configuration of design 1 (Omega-shape mechanism); (**b**) shows the configuration of design 2 (Fulcrum-shape mechanism); (**c**) shows the configuration of design 3 (Archway-shape mechanism); (**d**) shows the configuration of design 4 (Scissor-shape mechanism).

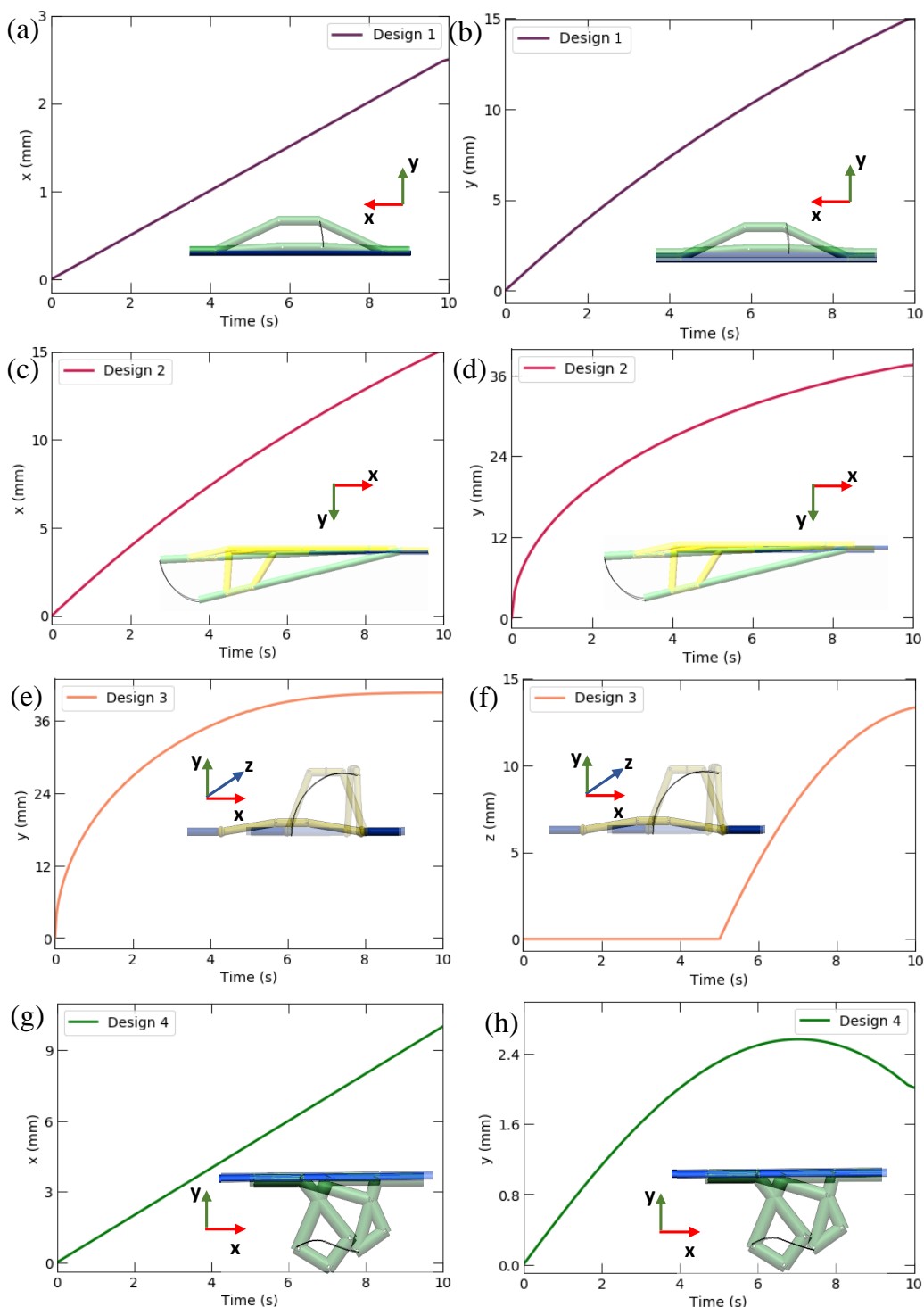

**Figure 4.** (**a**,**b**) The Omega-shape mechanism's displacement along the *x* direction and *y* direction respectively during the deployment process; (**c**,**d**) describe the Fulcrum-shape mechanism's displacement along the *x* direction and *y* direction respectively during the deployment process; (**e**,**f**) describe the Archway-shape mechanism's displacement along the *y* direction and *z* direction respectively during the deployment process; (**g**,**h**) describe the Scissor-shape mechanism's displacement along the *x* direction and *y* direction respectively during the deployment process.

## 2.1. Design 1: Omega-Shape Mechanism

The Omega-shape mechanism is designed as a slider-link mechanism configuration, and two 20 mm × 5 mm × 3 mm N38 NdFeB magnets are embedded and integrated with

straws. Two external magnets are placed on both sides to deploy this design. The actuation mechanism involves an external magnet anchoring link 1, while the other external magnet produces the linear translational motion of link 5 to deploy link 3 (Figure 3a). Another feature of the Omega-shape mechanism is that the external straw can add enough constraints, ensuring the stability of the final actuated state of this design. The stability is tested when the prototype retains its fully deployed position even after removing the external magnet anchoring link 1. In extreme cases, inadequate constraints result in losing control over the intra-abdominal robotic device, and surgical procedural complications can arise. Hence, the external straw reduces the external magnetic footprint on the abdominal surface during surgery to minimize the potential of undesirable magnetic coupling mentioned above.

### 2.2. Design 2: Fulcrum-Shape Mechanism

The Fulcrum-shape mechanism utilizes a closed-loop kinematic chain for actuation. Three 20 mm × 5 mm × 3 mm N38 magnets are embedded in this prototype design. The functions of magnet A are to make the prototype more compact and flat, making it easy to move the prototype to the desired position. The movement can be achieved by coupling an external magnet with Magnet A and another magnet with Magnet C (Figure 3b). When Design 2 is navigated to the desired position, we can deploy the mechanism by re-coupling the external magnet from Magnet A to Magnet B. Magnet B and C work in tandem for actuation during the deployment process. Magnet C acts as an anchor, while Magnet B translates linearly towards Magnet C.

### 2.3. Design 3: Archway-Shape Mechanism

An Archway-shape mechanism has four 20 mm × 5 mm × 3 mm N38 embedded magnets, and two magnets are placed at each end of the prototype (Figure 3c). To actuate the prototype, we first drag the two external straws and fully expose the incisions made on the inner straw. Then, we manipulate the prototype to achieve the position shown in Figure 3c. Finally, we drag links 1 and 5 closer together while the two links remain parallel to one another. Archway-shape mechanisms with three different parameters are manufactured, and the lengths of links 2, 3, and 4 of each prototype are (1) 3.5 cm, 3.0 cm, 3.5 cm, (2) 4.0 cm, 2.5 cm, 4.0 cm, and (3) 4.5 cm, 1.5 cm, 4.5 cm, respectively. We will test the impact of the lengths of links on the actuation success rate. In this design, the stopper system holds the entire prototype together by preventing the inner straw from falling out of the outer straws. A thin metal wire is attached to the inner straw, threaded through the incision made on the outer straw, and wrapped around the outer straw. The length of the incision extends to the point where it is enough to expose all the hinge joints on the inner straw. The thin metal wire allows the rotational motion of the inner straw when the prototype moves from the position, as shown in Figure 3c. Then, the fully actuated state can be attained.

### 2.4. Design 4: Scissor-Shape Mechanism

The Scissor-shape mechanism has two 20 mm × 5 mm × 3 mm N38 embedded magnets, as shown in Figure 3d. The design is a platform scissors lift mechanism. To actuate the design, the two magnets are simply moved towards each other. Furthermore, the Scissor-shape mechanism is modified to have curved incisions compared to Fulcrum-shape mechanisms with triangular incisions and Archway-shape mechanisms with triangular incisions. With the curved incision, the Scissor-shape mechanism can have a larger bending angle between its links than the Omega-shape design and Fulcrum-shape do. Meanwhile, the curved incisions provide more bending compliance compared to triangular incisions.

## 3. Results

This section records and consolidates the experimental observations of 4 designs (Omega-shape mechanism, Fulcrum-shape mechanism, Archway-shape mechanism, and Scissor-shape mechanism) regarding the most effective external driving magnet, actuation

success rate, workspace characterization, load tolerance, end effector force exertion, and the experimental results for the laser steering application.

### 3.1. Actuation Success Rate

The optimal set of external driving magnets for the deployment of each design is determined by testing various combinations of external magnets on each of the prototype designs while recording the actuation success. If the design fails to be fully deployed by any external driving magnets, they are deemed to be an unsuccessful design. As for the designs with the successful actuation, further analyses will be conducted on the prototype with the optimal performance, using their optimal set of external driving magnets.

As for the Omega-shape mechanism, the two units of magnets that optimally deploy this design are a stack of six 20 mm × 10 mm × 3 mm N35 NdFeB magnets and a single N52 cube 25 mm NdFeB magnet. The latter has a stronger magnetic remanence than the former. All prototypes are successfully actuated when the stronger N52 cube magnet is used as the actuating magnet. However, when the weaker unit of magnets is chosen for actuation, only the prototypes with lengths of links 2 and 4 above 4 cm are always successfully actuated (Figure 5a). Thus, we can conclude that prototypes with greater lengths of links 2 and 4 are more easily actuated. Figure 5b verifies these conclusions as it illustrates that the prototypes with longer lengths of links 2 and 4 require a shorter time to reach full actuation.

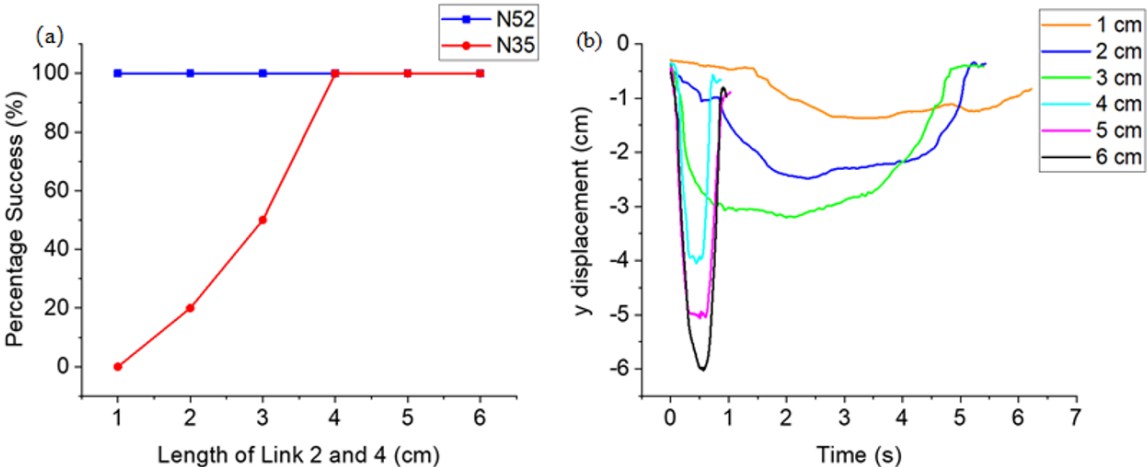

**Figure 5.** (**a**) Actuation Success Rate. (**b**) Actuation rate of various prototypes with varying lengths of links 2 and 4.

Considering the structure parameters of the Omega-shape mechanism, the prototype with links 2 and 4 at 4 cm gives the best actuation performance. This is evident from Figure 5a, as it can be actuated 100% of the time with the weaker N35 magnet, and it can also be fully actuated within the shortest time. The possibility of using a weaker N35 magnet for actuation is significantly beneficial as it reduces the likelihood of undesirable magnetic coupling between the external magnets. Hence, the prototype with links 2 and 4 with 4 cm will be used for further analysis.

With regard to the Fulcrum-shape mechanism, two units of magnets are utilized to actuate this design, namely an N52 cube 25 mm NdFeB magnet and a stack of five 20 mm × 10 mm × 2 mm N38 NdFeB magnets. The stack of magnets is used as the anchoring magnet to be coupled with Magnet C, while the cube magnet is used for actuation (Figure 3b). The set results in the actuation success rate at 100%.

All the prototypes of the Archway-shape mechanism with varying dimensions are all successfully actuated with negligible differences in their actuation time. The prototype with links 2, 3, and 4 at the length of 4.0 cm, 2.5 cm, and 4.0 cm will be used for further analysis. The Scissor-shape mechanism achieved a 100% actuation success rate using two

stacks of five 20 mm × 10 mm × 2 mm N38 NdFeB magnets. There is no external magnetic coupling during actuation.

### 3.2. Workspace Characterization

The workspace characterization is performed to show the effective workspace of each design. During actuation, the prototypes' motions are tracked from several crucial camera angles so that the motion analysis is considered in the three-dimensional space.

Figure 6a–d shows the trajectories and workspace of the four successful designs' external and internal components. Since all the designs have two external units of magnets, the workspace and trajectories are measured, with one external unit being an anchor while the other acts as the driving unit. The joint (e.g., link 2–3 of Omega-shape mechanism) and point (end effector of Fulcrum) used for workspace characterization is not only the position along with the prototype that has substantial displacement during deployment, but also the position along the prototype body where the external tools such as laser can be attached to.

Comparing the magnets' trajectories amongst the four designs above, the actuation process of the Archway-shape mechanism involves three steps. At the same time, Omega-shape, Fulcrum-shape, and Scissor-shape only require a single linear translational motion to actuate their prototype fully. The results also present that the Fulcrum-shape mechanism has the smallest range of motion amongst all four designs, as seen from the trajectory of its end effector compared to the trajectories of the other designs' joints. The end effector of the Fulcrum-shape mechanism can only be displaced to a maximum of about 2 cm in the *y*-axis, whereas the other designs can be minimally displaced to 3 cm in the *y*-axis.

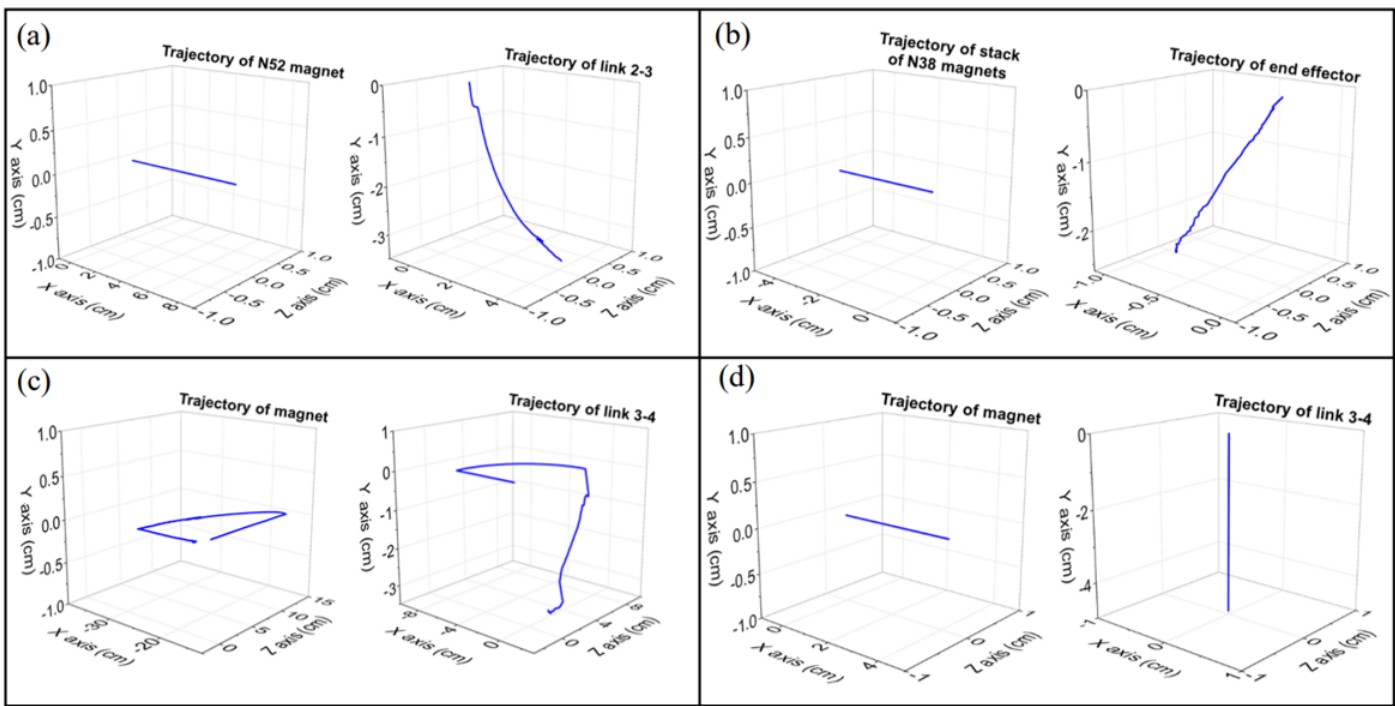

**Figure 6.** Workspace characterization plots showcasing the two-dimensional motion and three-dimensional reachable workspace by each of the deployable tubular mechanisms, as well as their respective external driving magnet. (**a**) Omega-shape mechanism; (**b**) Fulcrum-shape mechanism; (**c**) Archway-shape mechanism; (**d**) Scissor-shape mechanism.

### 3.3. Force Analysis

The force analysis presented in Figure 7a–d is conducted on the prototypes in their inactivated state. The load force is gradually applied to the end effector of the prototypes. The displacement of the prototype's end effector from the initial position is recorded.

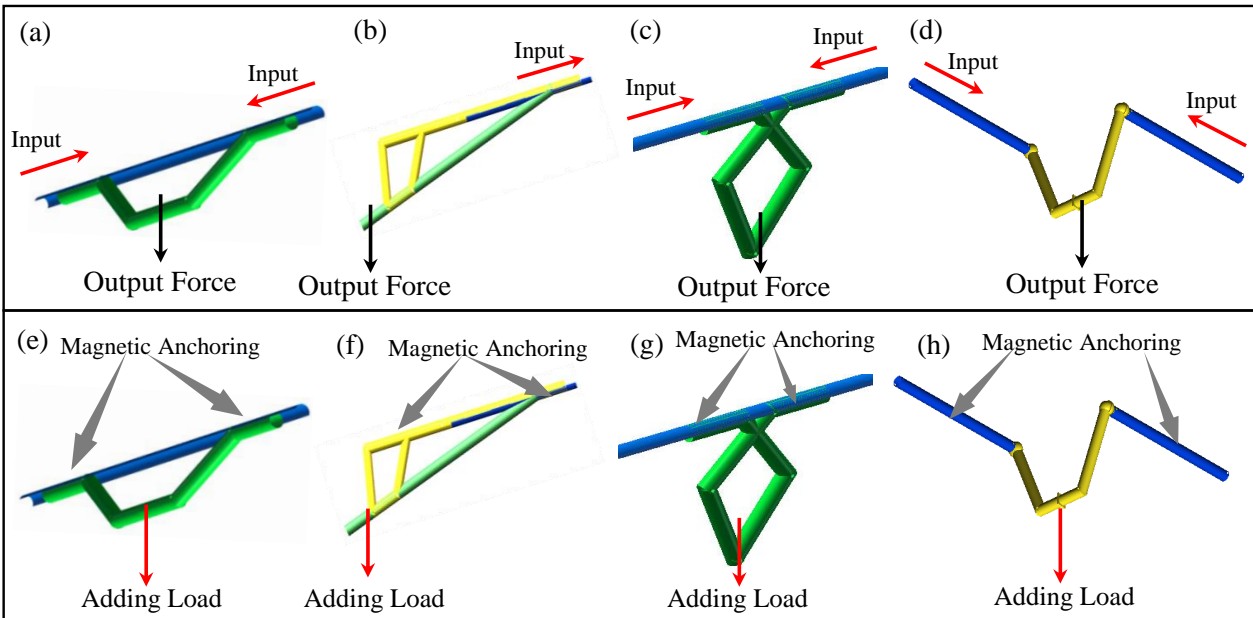

**Figure 7.** (**a**–**d**) The experiment setup for force analysis respectively corresponding to Design 1–4; (**e**–**h**) describe the experiment setup for load analysis.

Figure 8 above reports the force exerted by the end effector of each prototype during a full actuation cycle. The results showed that the Archway-shape mechanism exerts the greatest force at its end-effector of about 2.07 N, while the Omega-shape mechanism exerts the smallest force of about 0.9 N compared to the other designs. Omega-shape mechanism and Archway-shape mechanism have a similar force profile with a single peak. However, the force profile of the Fulcrum-shape mechanism is shown to have a great contrast with the other designs. The Archway-shape mechanism's force profile observed several small peaks before and after a single tallest peak.

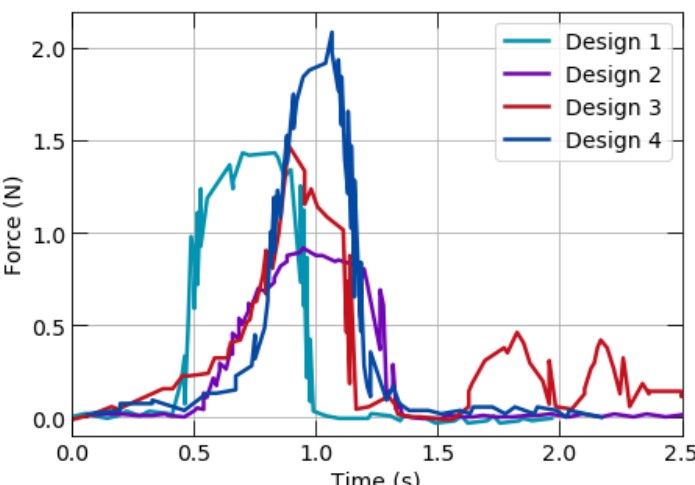

**Figure 8.** Force exerted by end effector during actuation among the Design 1 (Omega-shape mechanism), Design 2 (Fulcrum-shape mechanism), Design 3 (Archway-shape mechanism), and Design 4 (Scissor-shape mechanism).

### 3.4. Load Analysis

The slope in Figure 9a correlates with the resistance against being displaced, where a smaller slope corresponds to an increased resistance ability. Hence, the load capacity of the designs can be observed in Figure 9.

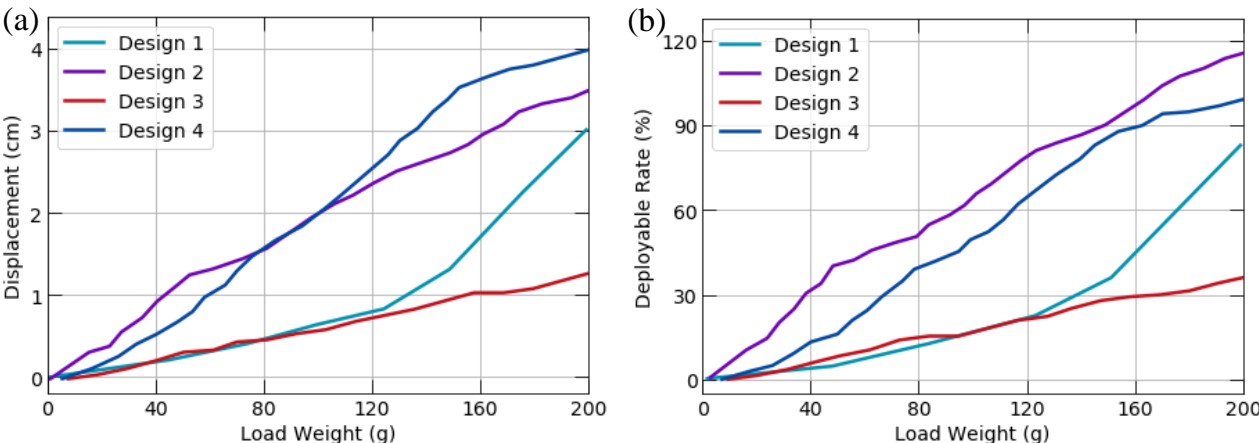

**Figure 9.** (**a**) Displacement and (**b**) deployable rates of all designs.

The Archway-shape design has the best tolerance for the load, as seen from its relatively low percentage displacement compared to the other designs. Omega-shape design experiences an exponential increase in its displacement with increased weights. This shows that Omega-shape Design has a greater stability when the load is less than 125 g, but becomes increasingly unstable when the load is high above 125 g. Scissor-shape design reaches 100% actuation state when the load is 200 g, while Fulcrum-shape design exceeds its 100% displacement as shown in Figure 9b with about a 115% percentage displacement under 200 g load.

Hence, it can be concluded that both Fulcrum-shape mechanism Design and Scissor-shape mechanism Design are relatively easily displaced when loads are attached to the prototypes.

### 3.5. Laser Steering Device

This section explains the first attempt to attach the laser device to the deployable tubular designs. This paper chooses a micro infrared light source with a maximum luminous flux below 100 lumens, maximum power of 1 W, and a non-tunable focus. This section reports four designs with an attached laser to demonstrate the trajectory tracking ability.

#### 3.5.1. Direct Steering via Macro Setup

The prototypes are attached with the laser light so that the deployment of the prototypes would bring about movement of the attached laser. A precision analysis is also performed on the laser application by measuring the deviation of the laser point from the desired path, which reflects the precision of the laser motion. The Tracker software extracts the deviation data from the video graph of the shape-tracing motion. The correlation between the laser point motion and prototype actuation is recorded and presented to reveal further the relationship between the laser steering capability and the deployable structure.

Direct steering refers to the laser beam being directly shone at the target plane. The steering is controlled by the actuation of the prototypes via manipulation of the external magnetic units. Figure 10 shows the movement of the end effector that creates the laser beam's vertical and horizontal motions. Since the feasibility of manual linear tracing is proven by manipulating the external magnetic unit to actuate the prototypes, the feasibility of tracing increasingly complex shapes was tested below.

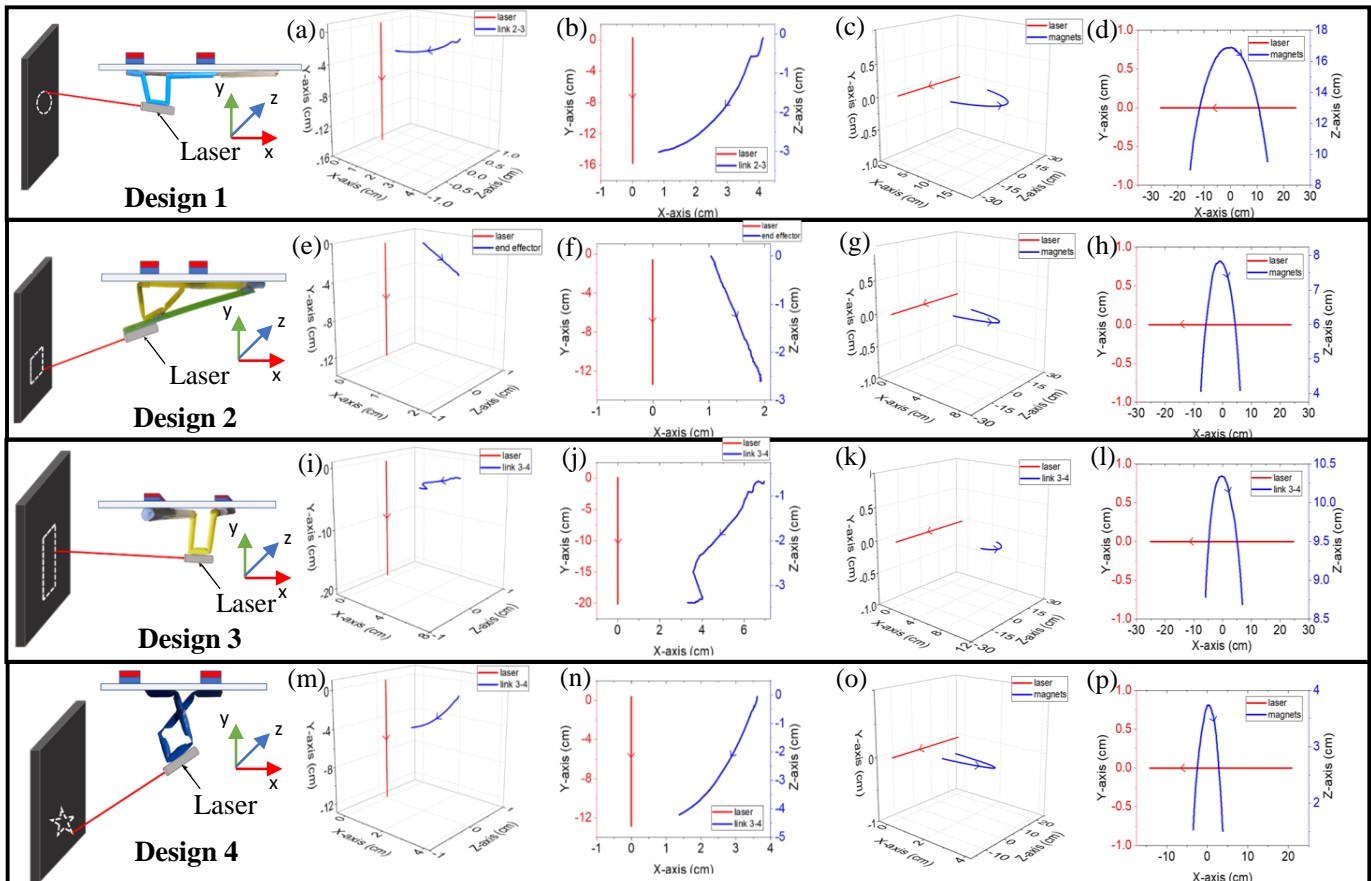

**Figure 10.** Plots illustrating the correlation of the motion of the laser-attached prototype designs with the vertical motion of the laser point. (**a–d**) Omega-shape mechanism; (**e–h**) Fulcrum-shape mechanism; (**i–l**) Archway-shape mechanism; (**m–p**) Scissor-shape mechanism.

Since all four designs can trace vertical and horizontal lines, the rectangle shape is tested next to explore their ability to change their tracing paths at an angle of 90°. Figure 11a–c shows that Designs 1 (Omega-shape mechanism), 3 (Archway-shape mechanism), and 4 (Scissor-shape mechanism) can decently trace rectangles of the respective dimensions. In contrast, Design 2 (Fulcrum-shape mechanism) failed to trace a rectangle. However, Design 2 (Fulcrum-shape mechanism) can only trace a rectangle with longer vertical edges.

Subsequently, the tracing of a square by the four designs tests the ability of the designs to trace a symmetrical shape. Figure 11d–f shows that Designs 1 (Omega-shape mechanism), 2 (Fulcrum-shape mechanism) and 4 (Scissor-shape mechanism) were able to perform the tracing of a square. Still, Design 3 (Archway-shape mechanism) was unable to do so. Moreover, Design 1 (Omega-shape mechanism) was able to perform tracing of a square with the largest dimension, followed by Design 4 (Scissor-shape mechanism) and Design 2 (Fulcrum-shape mechanism), respectively.

Next, the circle shape was successfully traced by Designs 1 (Omega-shape mechanism), 2 (Fulcrum-shape mechanism), and 4 (Scissor-shape mechanism), further confirming that these designs can decently trace symmetrical shapes. Additionally, Design 3 (Archway-shape mechanism) is proven to be unable to achieve symmetrical tracing as it cannot trace both a square and a circle.

Finally, tracing a complex shape, a star, was tested for all four designs. Only Designs 3 (Archway-shape mechanism) and 4 (Scissor-shape mechanism) were able to perform this tracing process with decent precision. The experiment video for direct steering section can be viewed in Supplementary Materials Video S1.

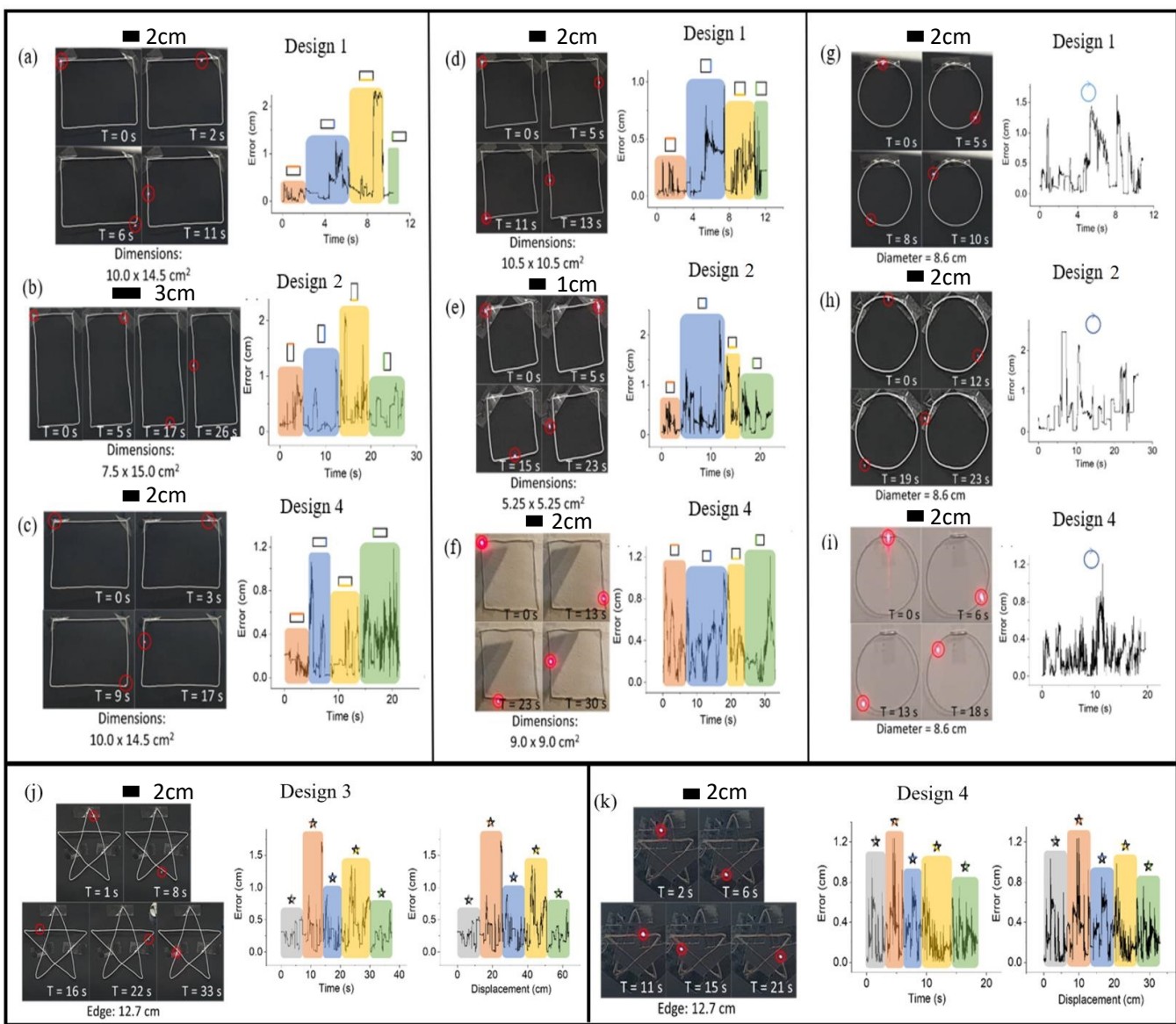

**Figure 11.** (**a**–**c**) The error of Designs 1 (Omega-shape mechanism), 3 (Archway-shape mechanism) and 4 (Scissor-shape mechanism) when tracing a rectangle; (**d**–**f**) illustrate the error of Designs 1 (Omega-shape mechanism), 2 (Fulcrum-shape mechanism) and 4 (Scissor-shape mechanism) when tracing a square; (**g**–**i**) illustrate the error of Designs 1 Omega-shape mechanism), 2 (Fulcrum-shape mechanism) and 4 (Scissor-shape mechanism) when tracing a circle; (**j**,**k**) illustrate the error of Designs 3 (Archway-shape mechanism) and 4 (Scissor-shape mechanism) when tracing a pentagram.

### 3.5.2. Indirect Steering via Macro-Mini Setup

Further improvements to the laser-assisted device have been attempted by introducing a mini component into the setup. The macro component consists of the prototype structure with the attached laser. Whereas, the mini component of the setup consists of two electromagnets, an additional embedded magnet, a relaying magnet coupled with the embedded magnet, and an iron sheet-attached mirror which was coupled with the relaying magnet (Figure 12). A DC voltage supply was connected to each of the electromagnets, and the voltage supplied to the electromagnets was manually controlled to vary the magnetic strength of each electromagnet. Electromagnet 1 will first be supplied a DC voltage of 12 V for 5 s. It will then be reduced to 0 V while the voltage supplied to electromagnet 1 will be increased from 0 V to 12 V for another 5 s. Hence, each cycle lasts for 10 s, and the cycle can be repeated for as long as desired. The purpose of varying the magnetic strength between

the two electromagnets is to attempt to manipulate the tilt angle of the attached mirror to steer the laser beam along the dotted line, as shown in Figure 12. This process of altering the supplied DC voltage was further improved via Arduino programming to automate the entire process.

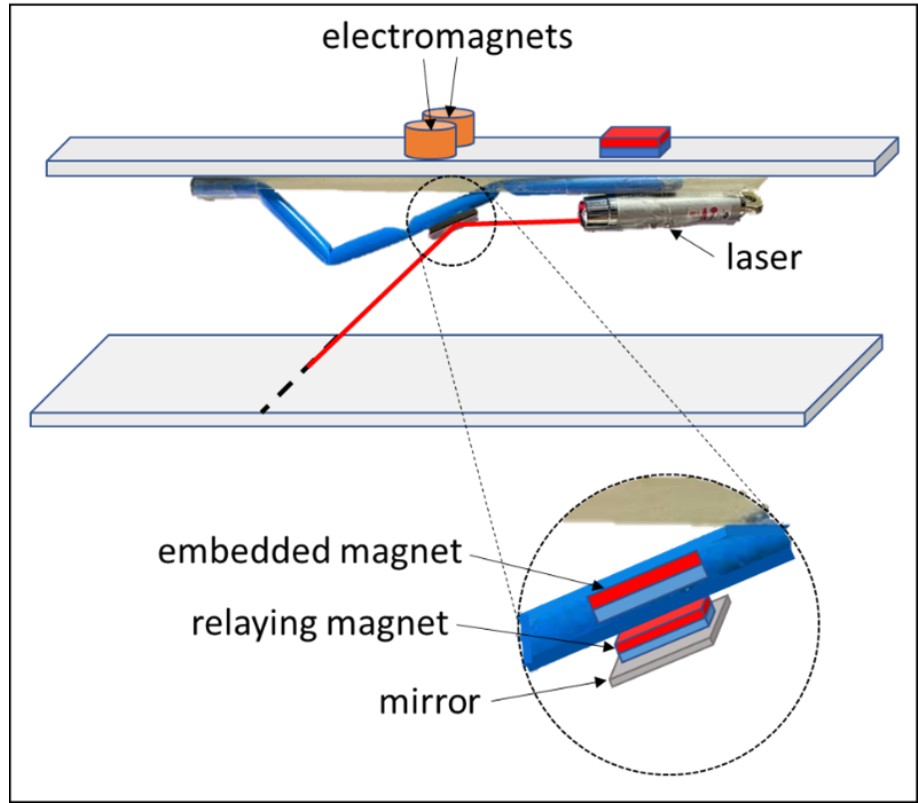

**Figure 12.** Schematic diagram of Omega-shape mechanism's macro-mini setup for indirect laser steering via manipulation of the mirror's tilt angle by alternating the electromagnetic strength.

After proving that the designs can trace various shapes, an attempt to improve the dexterity and precision of the designs was explored. The macro-mini setup involving indirect laser steering through a mirror reflection was designed. Electromagnets, instead of permanent magnets, were used to control the laser steering by manipulating the tilt angle of the mirror. The macro-mini setup successfully achieved a linear laser point displacement of $+/-$ 0.2 cm from its original position. This was successfully carried out by manually varying the voltage from the DC supply and using the Arduino programmed setup. The experiment video for indirect steering section can be viewed in Supplementary Materials Video S2.

## 4. Discussion

Four designs are proposed and experimentally shown to be successful in their actuation mechanism and their ability to perform the laser steering experiment. Then, we discuss the optimal design in terms of the experimental results above. The actuation mechanisms of Omega-shape and Fulcrum-shape designs are intuitive by manipulating a single external magnetic unit to achieve X-Y plane motion (Figure 6). Design 4 (Scissor-shape mechanism) can exert the greatest force at 2 N. Figure 8 shows that Design 3 (Archway-shape actuation mechanism) with the smaller force peaks reflects the longer time required by the design to complete an actuation cycle due to its multi-step actuation mechanisms. Therefore, the Archway-shape actuation mechanism will not be a good option for procedures that require a single, targeted force exertion with a fast reaction time. The results from the load analysis (Figure 9) have shown that Design 3 (Archway-shape mechanism) has the best

tolerance for load, and it is suitable to carry large-weight surgical instruments such as a retractor. In Figure 9b, Design 2 (Fulcrum-shape mechanism) exceeded 100% displacement and the prototype reached its maximum actuation displacement when only 160 g of the load was added. This implies that any subsequent load added to the prototype will cause its structure to stretch or bend beyond its maximum displacement (maximum displacement when 0 g of the load was added) to compensate for its limited range of motion. Hence, it is crucial to factor in the load tolerance of the designs and their range of motion when considering the applicability of the designs to be used as a retractor.

Each laser-attached prototype design was experimented with to test their ability to trace the shapes stated in Figure 13. This was done by manipulating the external units of permanent magnets. The types of shapes that each design can successfully trace are summarized in Figure 13. The mean tracing error reflects the tracing precision of each design, where a lower mean error corresponds to greater precision.

Design 4 (Scissor-shape mechanism) is shown to have the best tracing ability as it successfully traced all the shapes, and its mean tracing error is also relatively low, as shown in Figure 13. Design 1 (Omega-shape mechanism) ranks second in its intuitiveness, ease, and ability to trace the various shapes. Design 2 (Fulcrum-shape mechanism) can draw a square but not a rectangle, as the shorter edge of a rectangle is a challenge for such a design. This challenge is due to its limited range of motion and its lack of precise control, reflected by its relatively high mean tracing error in Figure 13.

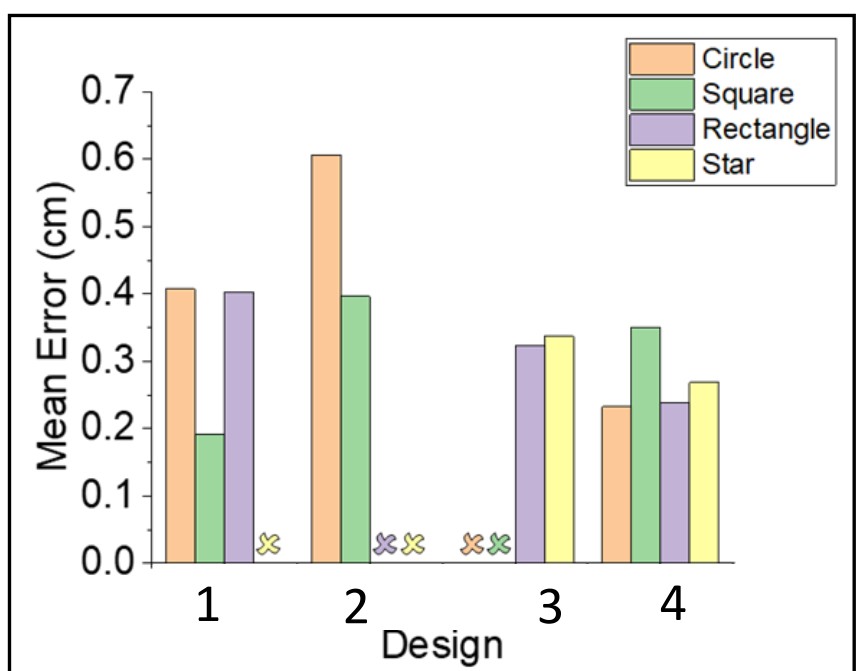

**Figure 13.** Mean tracing error of Designs 1 (Omega-shape mechanism), 2 (Fulcrum-shape mechanism), 3 (Archway-shape mechanism), and 4 (Scissor-shape mechanism) for various shapes. X means that design failed to trace the respective shape.

Furthermore, its limited range of motion is also a shortcoming that limits its ability to trace only shapes of smaller dimensions compared to the other designs. For example, the Fulcrum-shape mechanism can trace a square with a maximum dimension of $-5.25$ cm $\times$ 5.25 cm, while the other designs can trace a square with edges of double that length. Another important finding is that the Archway-shape mechanism is more suitable for tracing straight lines than curved lines, as seen from its failure to trace a circle. Archway-shape mechanism's results also interestingly oppose Design 2's, as it can trace a rectangle but not a square. This is due to its inability to precisely control the extent of displacement of the laser point along the vertical path while actuating Design 3 (Archway-shape mecha-

nism). Hence, the Archway-shape mechanism can only trace rectangles with the longer vertically orientated edges. This limitation reduces its ability for symmetrical tracing.

However, the current system does not consider the introduction of visual information in the current work. The light beam steering video (not in real time) is utilized as post-processing to track the laser point's location and then evaluate its performance offline. Therefore, these demonstrations aim to verify the proposed structure's feasibility. Yet, indeed, as mentioned, the current designs are all under "open-loop" control or manual control, so the reproducibility and robustness of such operation cannot be guaranteed in most cases. The introduction of visual feedback control will lead to a more accurate actuation than the current open-loop control, and the position error will become smaller through the feedback correction so that the position control of functional components (such as laser or light beam, endoscope, etc.) become more accurate and stable. Generally speaking, there are two ways to introduce and provide visual feedback at MAGS: one is a detached method, where the endoscope is considered as an extra functional component, such as rigid MAGS endoscope [38], soft-bodied MAGS endoscope [39]; and the other is an integrated introduction, where the endoscope is integrated into the deployable mechanism structure (relatively central position). Such a camera system based on the controlled PM position has been proposed with reliable motion and the fine-tuning of one of the DOFs used to change the field of view for a LESS scenario [40]. After that, the position of the laser point can then be tracked by the visual servo technique, thus enabling precise control of the deflection trajectory.

## 5. Conclusions

Four strategically fabricated deployable tubular structures (Omega-shape, Fulcrum-shape, Archway-shape mechanism, and Scissor-shape mechanism mechanisms) are presented to address the limitations of current NOTES and LESS procedures. Four proposed designs are proven to be successfully actuated and showed great potential in laser steering applications. The success demonstrated the opportunities for these structures and designs to be further developed.

The readily available information on the characterization of each design in this paper provides a basis for appropriate deductions on their applicability to new medical or surgical procedures, allowing embarkation towards a new direction in future works. With a macro-mini experimental setup, the challenge of utilizing a bulky and strong electromagnet can be overcome while reaping the benefits of the incorporation of an electromagnet. These benefits include the real-time variation of magnetic strength and reduced safety concerns.

By and large, the proof of concept of a magnetically tubular deployable structure with its potential applicability of laser steering in the application of laser-assisted devices is demonstrated in this paper. Compared with the other devices in the market, the proposed mechanisms introduced in this paper are proven to increase user intuitiveness, effectiveness, safety, and affordability. In contrast, the device's precision is more accurate than the hand-held treatment device. The deployable structure would be further miniaturized in the future, and magnetic actuation should be motorized. Furthermore, future research can adopt suitable biocompatible materials to manufacture these tubular structures.

**Supplementary Materials:** The following supporting information can be downloaded at: https://www.mdpi.com/article/10.3390/act11050124/s1, Video S1: Experiments of direct steering via macro setup. Video S2: Experiments of indirect steering via macro-mini setup.

**Author Contributions:** Conceptualization, H.R.; methodology, H.R., W.Y., R.T. and J.S.W.; validation, W.Y., R.T. and J.S.W.; investigation, W.Y., R.T. and J.S.W.; resources, H.R.; data curation, W.Y., R.T., J.S.W. and H.R.; writing—original draft preparation, W.Y., R.T., J.S.W. and H.R.; writing—review and editing, W.Y., R.T. and H.R.; visualization, W.Y., R.T. and J.S.W.; supervision, H.R.; project administration, H.R.; funding acquisition, H.R. All authors have read and agreed to the published version of the manuscript.

**Funding:** This work was supported by the National Key R&D Program of China under Grant 2018YFB1307700 (also with subprogram 2018YFB1307703) from the Ministry of Science and Technology (MOST) of China, Chinese University of Hong Kong (CUHK) Direct Grant 4055139 for the project on Multiphysics Study of Magnetically Deployable Robotic Collapsible Structures, Hong Kong Research Grants Council (RGC) Collaborative Research Fund (CRF C4063-18G and C4026-21GF).

**Institutional Review Board Statement:** Not applicable.

**Informed Consent Statement:** Not applicable.

**Data Availability Statement:** Not applicable.

**Conflicts of Interest:** The authors declare no conflict of interest.

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
