# Peer review of "Deployable Tubular Mechanisms Integrated with Magnetic Anchoring and Guidance System"

_actuators, doi:10.3390/act11050124_

Round 1
Reviewer 1 Report
Four novel deployable tubular mechanisms are proposed integrated with MAGS in this paper. Laser steering experiments were performed to evaluate the performance of the four proposed mechanisms. The proof of concept of the magnetically tubular deployable structures are useful attempts to address the limitations of current NOTES and LESS procedures.
Generally speaking, this paper is well-organized and addresses an important research topic on surgical robot. So I would recommend the paper to be accepted with minor revision. To improve the quality of this paper, the following issues should be addressed:
1) For figure 3, the x, y, z coordinates are not clear, please modify it. For figure 4, please mark a and b.
2) For figure 6(f), please check that there is only one magnetic anchoring?
3) For line 200, What does ?? stand for?
4) In section 2.5.1, "A precision analysis is also performed on the laser application by measuring the deviation of the laser point from the desired path along the edge of the various shapes", why use different shapes instead of the same shape, such as square or circle?
5) For figure 9, the x, y, z coordinates are not clear, please modify it.
6) Figure 10 is not clear, suggest relayout.
7) Some references should be modified to meet journal requirements, such as "Omisore, O. M.; Han, S.; Xiong, J.; Li, H.; Li, Z.;Wang, L. A review on flexible robotic systems for minimally invasive surgery. IEEE Transactions on Systems, Man, and Cybernetics: Systems 2020", there is no vol and pages.
Author Response
Dear Reviewer,
Many thanks for your advices! It is very helpful to promote the quality of the article.

Reviewer 2 Report
This paper presents foldable structures with permanent magnets embedded inside which could be actuated by external magnets. The foldable structures are made of a series of links where the folding could be pushed to a boundary wall and further be moved by manipulating external magnets. Four designs have been reported where with the hinge joints are at different positions and orientations along with the straws. The authors performed quantified analysis to discuss the deploying behavior of each design, the applied force and the resulted motion. The functions of the foldable structures are demonstrated to manipulate a laser spot where a laser pointer is embedded inside the link. Please find below the comments to help improve the paper.
- The motivation of using a foldable structure demonstrated in this work is weak. With a magnetic capsule endoscope, similar locomotion and function e.g. carrying a laser pointer could be realized. This paper didn’t justify clearly why a foldable structure is needed. In my opinion, at least the demonstration did not prove this point.
- The motion of the foldable structures is driven by the magnetic gradient pulling forces. There is friction applied on the foldable structures by the environment surfaces. How is the friction between the body links and the surface? How does the friction affect the structure motion?
- In Fig. 9, the actuation distance of the external magnets looks very small. How does it affect the practical applications? For example, in the abnormal applications, what are the typical distance from the skin surface to the GI tract cavity? These data should be included and compared.
- In Figure 3, what do the red and blue colors indicate? It would be helpful to mark the external magnet position and orientation in each subfigure. The force and torque should also be included to help illustrate the actuation.
- Scale bars should be included in all experimental figures.
- The authors showed four designs. However, it is unclear which one is the optimal design. It is also unclear how a specific design should be selected in a given application scenario.
- The authors mentioned scaling down the size of the foldable structure in the future. How is the scalability of the system? A scaling analysis would be helpful to help understand the limitations of this work. Are there significant interactions between the magnets embedded in different links?
- This work failed to cite many important works in magnetically actuated robots for minimally invasive medical applications, such as miniature soft magnetic robots, magnetic endoscope capsule robots, and magnetic continuum robots.
- A video is suggested to be included to show the experimental results.
Author Response

(The authors gave the same response as above.)

Reviewer 3 Report
The authors presented pipe-based 4 designs with a permanent magnet or electromagnet-based remote manipulations. Depending on the robot design and permanent magnets, the robots acquire certain poses. There have been added electromagnetic components to further advance the external magnetic fields generated. My major comments are below:
- The paper is showing combining pipes with some magnets and creating 4 shapes that seem like working enough. Eventhough authors claim this is novel, for such an application base study, it is not enough to make something novel but also suboptimal. The paper creates the impression that the authors found out some interesting enough combinations while playing with magnets. The scientific approach to this should start with a design rubric and design methodology. By sticking to the methodological rules, the authors should come up with 1 or multiple designs. There is no point on creating these 4 random shapes and naming them while there is no real counterpart of those shapes in the application. Every design naturally provides different results but without knowing why these 4 designs are being chosen, such a comparison does not teach much to the readers.
2. There is no explanation on how magnetic field actuation works. What kind of forces acting on there, torques, forces? It is described in very unofficial words. To the readers, it sounds like we merged something and this magic has happened. Readers have no idea how this has happened, why did you integrate those magnets, how did you select the magnet sizes etc (quantitatively, not qualitatively with vague words). Similarly, for external magnetic fields, why did you choose those magnets, shapes etc. It seems too ad-hoc. The magnetic science is extremely weak.
3. Discussion part has so much new information at it is so long.
4. Paper is written without a scientific language. The paper sounds like an interview conversation with the authors throughout the paper.
5. Many English grammar mistakes and too many vague nonscientific words are being used throughout the paper.
Author Response

(The authors gave the same response as above.)

Round 2
Reviewer 2 Report
The authors have addressed most of my comments. Please find one extra comment to be clarified.
What kind of medical applications do the authors target when the robot can only locomote on the abdominal wall? If it is in the GI tract, the actuation distance of the external magnet should be much more than 0.7 cm, isn’t it? Please comment the practicability of the method in terms of the actuation distance.
Reviewer 3 Report
The authors improved the paper in the direction of my comments successfully. The paper is in a nice shape and more cohesive, however, it is lacking a few more points to be improved. I appreciate the work authors put through and improved the quality of the paper. The figures are clear and the authors added a new figure describing the mechanism. Please see my new comments:
Major Comments:
1. Regarding the magnetic robotics community, one critical issue with such devices is imaging feedback. Once the robot is inside the human body, we can not see its orientation or shape since there is no direct line of sight. How do the authors think that they can know their novel design would ensure a high-performance rate? It would be good to extend the discussion with the dimension of the noninvasive imaging techniques to monitor these straw-magnet mechanisms. Do they think open-loop changing the currents or permanent magnets would always provide the same result? Do they need an imaging method and combine it with closed-loop autonomous control algorithms? Please improve the discussion section along these lines.
2. Orientation-based guidance and steering have been one of the focuses in the magnetic robotics community. The literature search is weak and leaves the readers in the dark based on what has been accomplished specifically at the steering and guidance level. Line 8 to 11 describes what has been done in the magnetical realm but the examples don't cover the true state-of-the-art in terms of width and breadth. The following closely related works should be added to explain how the magnetic robots are being used to implement 'orientation-dependent' tasks such as laser guiding, biopsy, hyperthermia, and drug delivery, and needle guiding.
For laser guiding example:
P1. Xu, T., Zhang, J., Salehizadeh, M., Onaizah, O., & Diller, E. (2019). Millimeter-scale flexible robots with programmable three-dimensional magnetization and motions. Science Robotics, 4(29), eaav4494.
For biopsy example:
P2. Hoang, M. C., Le, V. H., Nguyen, K. T., Nguyen, V. D., Kim, J., Choi, E., ... & Kim, C. S. (2020). A robotic biopsy endoscope with magnetic 5-DOF locomotion and a retractable biopsy punch. Micromachines, 11(1), 98.
For camera pointing example:
P3. Erin, O., Alici, C., & Sitti, M. (2021). Design, actuation, and control of an MRI-powered untethered robot for wireless capsule endoscopy. IEEE Robotics and Automation Letters, 6(3), 6000-6007.
For drug delivery and hyperthermia example:
P4. Erin, O., Boyvat, M., Lazovic, J., Tiryaki, M. E., & Sitti, M. (2021). Wireless MRI‐Powered Reversible Orientation‐Locking Capsule Robot. Advanced Science, 8(13), 2100463.
For needle insertion example:
P6. Vartholomeos, P., Qin, L., & Dupont, P. E. (2011, September). MRI-powered actuators for robotic interventions. In 2011 IEEE/RSJ International Conference on Intelligent Robots and Systems (pp. 4508-4515). IEEE.
Minor Comments:
1. Line 79 instead of * sign use x when referring to dimensions.
2. the units and values should have 1 space between them. I see inconsistent usage. in some parts, it is referred to as 3.2cm. It should be '3.2 cm' with the space between.
3. Even though the manuscript is in much better shape in terms of English fluency and typos, various punctuation errors are still present in the manuscript. consistent spacing and proofreading are needed.
4. Line 310 should start with a capitalized letter.
4. The abbreviation NdFeB is being used before it is explicitly explained. NdFeB abbreviation should be explicitly written at its first use in the manuscript.
5. Throughout the manuscript, the authors used many different types of magnets. Please provide vendor and model information for all types of magnets mentioned in the manuscript.
